# Street Food in Malaysia: What Are the Sodium Levels?

**DOI:** 10.3390/foods11233791

**Published:** 2022-11-24

**Authors:** Hasnah Haron, Zainorain Natasha Zainal Arifen, Suzana Shahar, Hamdan Mohamad, Siti Farrah Zaidah Mohd Yazid, Viola Michael, Rabindra Abeyasinghe, Tanaka Taketo, Kathy Trieu

**Affiliations:** 1Nutritional Sciences Programme, Centre for Healthy Ageing and Wellness (H-Care), Faculty of Health Sciences, Universiti Kebangsaan Malaysia, Kuala Lumpur 50300, Malaysia; 2Dietetic Programme, Centre for Healthy Ageing and Wellness (H-Care), Faculty of Health Sciences, Universiti Kebangsaan Malaysia, Kuala Lumpur 50300, Malaysia; 3Non-Communicable Disease Section, Disease Control Division, Ministry of Health, Malaysia, Putrajaya 62584, Malaysia; 4Enforcement Section, Allied Health Sciences Division, Ministry of Health, Malaysia, Putrajaya 62050, Malaysia; 5Representative Office for Malaysia, Brunei Darussalam, and Singapore, World Health Organization, Cyberjaya 63000, Malaysia; 6The George Institute for Global Health, Newtown, NSW 2042, Australia

**Keywords:** street food, sodium content, main meal, snack, dessert, Malaysia

## Abstract

Street food is a major source of food in middle- and low-income countries as it is highly accessible and inexpensive. However, it is usually perceived as unhealthy due to the high levels of sodium, sugar, and fat content. However, there is little analytical data on the sodium levels in the street foods of Malaysia. This study started with a survey to determine the most frequently available street foods in every state in Malaysia, followed by food sampling and the analysis of sodium (reported mg/100 g sample). Street food in the snack category contained the highest amount of sodium (433 mg), followed by main meals (336.5 mg) and desserts (168 mg). Approximately 30% of the local street food in this study was deep-fried. Snacks from processed food (8%) contained high sodium content (500–815 mg). Fried noodles and noodle soup contained the highest amount of sodium (>2000 mg sodium) based on per serving. Most main dishes that use a variety of sauces contained high amounts of sodium. These findings were recorded in the Malaysian Food Composition Database. Moreover, this study could raise awareness and serve as baseline data for future interventions on the sodium content in the street foods of Malaysia.

## 1. Introduction

According to the World Health Organization Report [1], non-communicable diseases (NCD) account for more than 70% of all deaths globally. Additionally, the number of NCD deaths is rising, and is expected to reach 52 million by 2030 [2]. In Malaysia, approximately 85,000 deaths were due to NCDs [1]. The Malaysian Community Salt Study [3], a population-based survey, reported that Malaysian adults consume a high amount of salt daily (7.9 g/day). This amount exceeds the recommended maximum salt intake of 5 g/day by the World Health Organization [4], resulting in hypertension, which affects one in three Malaysian adults [5]. Hypertension increases the risk of stroke, heart disease, and chronic kidney disease [6,7,8]. 

Due to the high burden of disease associated with excess salt intake, the Ministry of Health Malaysia (MOH) developed a national salt reduction strategy in 2010, to reduce daily salt intake by 30% by 2025 [9]. A proposed approach is the monitoring of sodium content in foods, which is then recorded in the available Malaysian Food Composition Database (MyFCD). This information is critical for food and health education, food research, and change monitoring. Since the implementation of the salt reduction strategy, sodium intake is monitored from the nutrition labels of packaged foods [9]. Nevertheless, little is known about the sodium content in local street foods [10]. This information is important because street foods are an increasingly common food source due to their accessibility, convenience, low cost, availability, and wide range of options [11]. Thus, this study aimed to determine the most frequently available street foods in every state in Malaysia and their sodium content. This data will be useful in developing effective strategies to reduce the salt content in local street foods.

## 2. Materials and Methods

Malaysia comprises 13 states and 3 federal territories (Kuala Lumpur, Labuan, and Putrajaya). The total population of Malaysia in 2022 is estimated at 32.7 million as compared to 32.6 million in 2021, with an annual population growth rate of 0.2%. The highest percentage distribution by ethnic group is Bumiputra (69.9%), followed by Chinese (22.8%), Indians (6.6%), and others (0.7%) [12]. In 2021, 77.7% of Malaysia’s total population lived in urban areas and cities [13]. There is currently no specific legislation for street foods and their sodium content. Street food stalls are randomly distributed throughout the cities and villages. 

Data collection was carried out in two phases. Phase I was a survey to determine the most frequently available street food in all 13 states and the federal territory of Kuala Lumpur. This survey started in early 2020 but was stopped due to a series of nationwide lockdowns. The survey resumed in December 2020 when the government allowed interstate travel once more. This was followed by Phase II, where street foods in all states of Malaysia were sampled from May to December 2021. The analysis of the sodium content in all samples started in early 2022. The flowchart of the whole study is illustrated in Appendix A. The procedures in this study were approved by the Research Ethics Committee of the National University of Malaysia with reference number UKM PPI/111/8/JEP-2020-433.

### 2.1. Phase I: Survey of Street Foods in All States of Malaysia

Phase I of this study was a field survey of locally available street foods conducted in 13 states and the federal territory of Kuala Lumpur in Malaysia. Street food is described as ready-to-eat food and beverages made and sold by vendors and hawkers, especially in the street and other similar public places. Based on the definition by the Food & Agriculture Organization (FAO) [14], street food can be distinguished from formal food service operations (e.g., cafes and restaurants) and includes foods from pushcarts, bicycles, baskets, balancing poles, or stalls without four permanent walls. The criteria for the street food stall to be selected in the study was one that did not have a fixed building or was confined within four walls (based on the definition by the FAO). This included individual stalls and stalls in the day and night markets. For each state, the operational night and morning markets were identified from the city council websites. A few night or morning markets in every state were surveyed.

A survey form (Appendix A) was used to record information about all street foods available from each location, such as the state, district, name of the street food, category of the street food (e.g., main meal, snack, or dessert), and the preparation method. The total frequency of each street food surveyed in each state was determined. The top 15 most frequently available street foods among each of the categories for every state were then identified and listed. The street foods selected were also based on the expected sodium content (e.g., desserts composed mostly of sugar were not included).

### 2.2. Phase II: Food Sampling 

#### 2.2.1. Sampling the Three Categories of Street Food Samples for Each State

A total of 15 frequently available street foods (7 main meals, 5 snacks, and 3 desserts), determined from the survey in Phase I, were sampled from every state. A total of 210 street food samples from all 14 states in Malaysia were analyzed for their sodium contents. The sampling of the street food was carried out according to a local method [15]. Each type of street food selected for sodium analysis was purchased from two different stalls within the same state. Since food sampling was carried out during the COVID-19 pandemic, the operating hours of street food stalls were affected and some of the stalls ceased operation. Thus, this study was limited by the availability of food. After purchasing the street food samples, they were stored in an ice box and transported (5 °C for 1.5 h) to the laboratories to protect the food from spoilage.

#### 2.2.2. Preparation of Street Food Samples for Sodium Analysis 

Preparation and analysis of street food samples were conducted in the food analysis laboratory based on the method in [15]. Each sample was weighed together with the packaging still intact using the top pan balance. Inedible portions (e.g., bones) were identified and removed. Following this, the same type of street food that was purchased from two different stalls was mixed and homogenized in a food processor. The homogenized samples were then kept in an airtight container and stored in a −20 ℃ freezer, before preparation for sodium analysis. Approximately 1–3 g samples were weighed into a 50 mL polypropylene tube, and 30 mL of 30% nitric acid was added into the tube for wet digestion. The closed tube was placed on a hot block at 95 °C for 1.5 h. The tube was then cooled to room temperature and topped up to 50 mL with deionized water. The stock standard used for sodium analysis had a concentration of 10,000 ppm. A series of intermediate standards (i.e., 0.1, 0.5, 1.0, 2.0, 5.0, 10, 20, 30, 50, and 100 ppm) were prepared from the stock standard. Each intermediate standard was prepared in a 1 L volumetric flask with diluted nitric acid. The sodium content was then analyzed in duplicates using inductively coupled plasma–optical emission spectrometry (Agilent Technologies, Santa Clara, CA, USA, 5100 ICP-OES) based on the protocol in [16]. Each sample was analyzed in duplicate and reported as mean ± standard deviation. The LOD and LOQ of this instrument were 0.0454 and 0.1610 mg/kg, respectively.

#### 2.2.3. Sodium Content Classification

There is no classification standard for high sodium ready-to-eat foods (i.e., street foods) in Malaysia. The sodium classification in Malaysia is meant for packaged foods with nutritional labeling. These packaged foods can be classified as low sodium (<120 mg/100 g sample), very low sodium (<40 mg/100 g sample), or sodium-free (5 mg/100 g sample). Hence, this study utilized the UK’s traffic light labeling scheme for classifying low sodium (<120 mg/100 g sample), medium sodium (between 120 and 600 mg/100 g sample), and high sodium (>600 mg/100 g sample) foods [17]. This classification is in line with the classifications used by the Food and Drugs (Composition and Labeling) Regulations in Hong Kong [18] and China (GB/T 23789-2009) [19].

### 2.3. Statistical Analyses

Here, IBM (Armonk, NY, USA) Statistical Package for Social Sciences (SPSS) version 25.0 was used to analyze the data. For Phase I, a descriptive test was performed to analyze the 15 most common street foods in Malaysia and sort them by food categories. The test included the preparation method, type, and category of the selected street foods. For Phase II, a descriptive test was used to analyze the average sodium content found in the selected street foods. Inferential statistical analysis, such as the one-way ANOVA test, was used to compare the average sodium content between the selected foods according to the food category and group. If the food groups and categories had unequal sample sizes and both one-way ANOVA and homogeneity of variance were significant, a Games–Howell post hoc test was performed to identify the specific differences between the three groups and categories.

## 3. Results

A total of 68 districts with 380 street food locations were surveyed. The highest number of locations surveyed (184 locations) was in Johor, whereas the lowest (4 locations) was in Kelantan. Among the total 10,520 types of street food surveyed in all states of Malaysia, the most common category of street food was snacks (40%), followed by main meals (37%), and then desserts (23%) (Table 1). Based on the survey, the main cooking method for street foods was deep-frying (28%), followed by steaming (13.3%), pan frying (11.1%), boiling (10.5%), grilling (9.4%), stir-frying (7.3%), stewing (6.1%), fermenting (5.3%), baking (4.9%), simmering (2.3%), roasting (9.4%), braising (9.4%), smoking (0.1%), and blanching (0.1%) 

A total of 210 samples were analyzed. There were 41 types of street foods from more than 1 state and 53 types of street foods from 1 particular state. The sodium content of similar street foods was averaged. The study reported the average sodium content in all 94 types of street foods as being in the range of 3.9–815.0 mg/100 g sample. Explanation on each street food can be found in Appendix A.

The average sodium content of street foods between the East Coast and West Coast of Malaysia was compared based on the food categories (21 and 63 main meals, 15 and 42 snacks, and 9 and 27 desserts, respectively). The sodium content in all three food categories between the two coasts was not significantly different (*p* > 0.05). On both coasts, snacks had the highest sodium content (411–506 mg/100 g sample), followed by main meals (327–392 mg/100 g sample), and then desserts (186–191 mg/100 g sample). 

The average sodium content of street foods between West and East Malaysia was compared based on the food categories (84 and 14 main meals, 57 and 12 snacks, and 36 and 7 desserts, respectively). The average sodium content in the snack category for both West and East Malaysia had the highest sodium content (450–481 mg/100 g sample), followed by main meals (376–404 mg/100 g sample), and then desserts (92–190 mg/100 g sample). The sodium content in the snacks and main meal categories between the two regions was not significantly different (*p* > 0.05). However, only desserts in West Malaysia reported significantly higher sodium content (190 mg/100 g sample) as compared to East Malaysia (92 mg/100 g sample) (*p* < 0.05).

Table 2 displays the average sodium content for each food category and food group. For food categories, snacks contained the highest average amount of sodium, followed by main meals and desserts. Both main meals and snacks contained significantly higher sodium contents when compared to desserts (*p* < 0.05). For food groups, processed foods contained the highest average amount of sodium compared to cooked dishes and local cakes. Most of the street foods were cooked dishes (63%).

Table 3 shows the sodium content in processed foods prepared as street foods. Only 8% of the street foods were prepared using processed foods, and most of the processed foods had medium to high sodium contents (451–815 mg sodium/100 g sample). Fried fish balls, fried chicken with cheese, fried crab meatballs, fried sausages, and fried chicken balls were categorized as processed foods prepared as street foods that contain high amounts of sodium (605–815 mg/ 100 g sample). The other street food samples in the snack category with high sodium content were fish-based snacks or keropok lekor (780.4 ± 109.5 mg/100 g sample) and seaweed pickles (761.0 ± 42.4 mg/100 g sample).

Table 4 shows that 46% of the main meal category consisted of noodles and rice, such as fried noodles, noodles with gravy, fried rice, and other cooked rice, which contained medium to high amounts of sodium. Noodle-based dishes, such as *soto*, fried noodles, *bakso*, and noodle soup contained more than 2000 mg sodium per serving. Fried kuey teow, *char kuey teow*, and noodles with curry or soy sauce gravy contained almost 2000 mg sodium per serving. *Laksa* (rice-based noodles with gravy made from fish) from Penang and Perak also contained almost 2000 mg sodium per serving.

Table 5 displays the rest of the street foods that contained medium sodium content (121.5–586.5 mg/100 g sample). This included 19 snacks, 15 desserts, and 7 main meals. There were 17 low sodium street foods (<120 mg/100 g sample) in this study. Most of the low sodium street foods (82.4%) were in the dessert category (Table 5).

## 4. Discussion

The highest percentage of street food surveyed in this study was from Selangor (22%), followed by Kuala Lumpur (20%). This could be attributed to Selangor having the highest population in 2022 [12]. Furthermore, Selangor and Kuala Lumpur were surveyed before the first nationwide lockdown in March 2021, when most street food stalls were still operating as usual. Main meals (37%) and snacks (40%) were the most common street food categories available in this study. This was similar to another study in Harare, Zimbabwe, where main meals (70.8%) and snacks (20.8%) were the most common [20]. Almost half of the main meals sampled in this study consisted of rice or noodle dishes. Rice is the staple food for half the world’s population, and it is cultivated predominantly in Asia [21].

The majority of street foods in this study were cooked foods (63%), and deep-frying was the most common street food preparation method. Deep-fat frying may be defined as immersing foodstuff in edible oil or fat at 150–200 °C [22]. Deep-fat frying is one of the most well-accepted methods in both local and international food preparation because of its convenience and highly stimulating properties (e.g., aroma and taste) to consumers [23]. 

There was no difference in the sodium contents of main meals and snacks between the different coasts or parts of Malaysia. This could be due to the same preparation methods being used for the common street foods. In the present study, 41 types of street foods were found across Malaysia. *Nasi lemak*, fried vermicelli, fried *mihun*, and fried chicken were frequently available across 12, 11, and 10 of the surveyed states, respectively. This was in line with a previous study, where *nasi lemak* and fried *bihun* were among the top 10 most consumed breakfast foods among adults in Malaysia [24].

Street foods with high amounts of sodium (605–805 mg/100 g sample) included processed foods, such as sausages, fish balls, chicken balls, and crab meat. The highest sodium content was found in fish balls. Fish balls are made from processed fish and can be eaten alone (fried) or in soups (boiled). The high sodium content in commercial fish balls accounts for the additional salt or sodium polyphosphate in preservatives and flavor enhancers [25]. Processed foods made from fish and chicken have sodium levels in the range of 479–500 mg/100 g sample, as listed on the nutrition labels [26]. 

The other street food in the snack category (452–584 mg sodium/100 g sample) included processed foods, such as beef burgers, nuggets, and sausages with cheese. *Keropok lekor* contained the second-highest sodium content in the snack category. It is a traditional fish-based snack from Terengganu and is popular on the East Coast of Peninsular Malaysia. *Keropok lekor* is commonly sold in all states of Malaysia, especially at roadside stalls and night markets [27]. In Phase I of this study, *keropok lekor* was frequently available in seven states, such as Johor, Sabah, Sarawak, Federal Territory of Kuala Lumpur, Negeri Sembilan, Terengganu, and Pahang. It is made from a mixture of minced/processed fish, sago flour, salt, monosodium glutamate, sugar, and ice-cold water [28]. The main source of sodium in the making of *keropok lekor* are salt and monosodium glutamate [29]. 

Seaweed pickle is very popular among the Bajau, and it was only collected in Sabah. Since it is harvested from the sea, the sea salt resulted in the high sodium content of the seaweed pickle. Normally, the seaweed pickle is served as a side dish to be taken with rice. Fried chicken with cheese was the fourth highest street food with a sodium content in this range. The average sodium content in 112 cheese products sold in major supermarkets in Malaysia was reported to be relatively high (856.54 mg/100 g sample) [26]. 

Fried noodles contained a high sodium content due to the ingredients used in the preparation stage. Fried noodles are often cooked with many sauces, especially soy sauce, thick soy sauce, oyster sauce, and chili sauce which contribute to high sodium content [30,31,32]. The sauces were added to add more flavor to the food and to make it more palatable to eat [33,34]. Additionally, the noodles contain a relatively high amount of salt in themselves [35,36]. *Kolo mee* or *mi kolok* is a signature dish from Sarawak. It has a high sodium content, which may be attributed to the sweet and salty soy sauce used in the dish. In Malaysia, the addition of salt and salty sauces to foods has been identified as the major source of sodium in the Malaysian diet [37].

In this study, 35% of main meals contained sodium content exceeding the recommended intake of 2000 mg sodium/day. However, the percentage is lower compared to the 62.6% of main meal dishes in 192 restaurants in China [38]. *Soto* is mainly composed of broth, meat, and vegetables, with a high sodium content (>3000 mg) per serving. Other noodle-based dishes, such as *bakso* (noodles with meatballs), noodle soup, fried *kuey teow*, and *char kuey teow* also contained high amounts of sodium (2000–2500 mg) per serving. This could be linked to the sauces (e.g., soy sauce and oyster sauce) used in their preparation. Soy sauce has been reported as one of the main sources of sodium during the preparation of ready-to-eat dishes in China [38]. *Laksa* uses fish-based broth that contains 1800–1900 mg sodium per serving, which could also be due to the use of shrimp paste.

Other rice-based dishes, such as *nasi tomato* (rice cooked with tomato paste), contained the highest sodium content, followed by *nasi minyak* (rice cooked with ghee and other condiments) and *nasi lemak* (rice cooked with coconut milk) with fried chicken (1400–1900 mg/serving). A normal set of *nasi lemak* (rice and condiment only) contained 643 mg sodium per serving. *Nasi kerabu*, a blue-colored rice dish that is eaten with dried fish or fried chicken (a source of sodium) and crackers (another source of sodium), contained 1000 mg of sodium per serving. 

In the dessert category, *apam balik* contained medium sodium content. *Apam balik* is a popular Malaysian traditional cake and is included as a Malaysian Heritage Food [39,40]. *Apam balik*, *apam balik* with egg, and *apam balik* with cheese contained 581, 600, and 1343 mg sodium per serving, respectively. The sodium content is directly associated with the use of sodium bicarbonate in its preparation. The addition of cheese to the snack and dessert categories increased the sodium content to the range of 35–55%. 

Street foods with a sodium content of less than 120 mg sodium/100 g sample were found mainly in the dessert category. Kamaruzaman et al. stated that as many as 70 types of traditional Malay cakes are still popularized in Malaysia [41]. The basic ingredients used to produce traditional sweet cakes are sugar, coconut milk, brown sugar, and Malacca sugar [42]. Thus, not as much salt is used in dessert preparation as compared to main meals and snacks. Snacks that are popular among the Chinese population, such as *chee cheong fun* (rice noodle roll) and *beh hua chee* (fried dough), also fall in this range (120 mg sodium/100 g sample).

The data on the sodium content in street food can be updated in the Malaysian Food Composition Database (MyFCD). This ensures that the nutrients contained in these street foods are known to the public, and that healthier food selections can be made. Apart from that, these findings justify the importance of salt reduction during the preparation of street foods and look into the use of salt substitutes, such as potassium chloride. Studies have demonstrated that replacing salt with sodium-reduced and potassium-enriched salt substitutes is better for the general population [43]. This study can serve as a foundation for future related studies.

## 5. Conclusions

Local street food in the snack and main meal categories contained significantly higher amounts of sodium than in dessert foods. Main meal street foods (e.g., noodle soup and fried noodles) and processed foods that were used in the preparation of street foods contain high amounts of sodium per serving, which exceeded the recommended daily sodium intake. *Keropok lekor* is one of the high-salt snack which is available in most states that should be targeted for reformulation and monitored over time. It is vital to disseminate information on the high sodium content of local street food to the public through advertisements and social media. The public needs to be informed about the sodium content in a single serving of street food to limit their sodium intake to 2000 mg daily. This practice will help reduce the sodium consumption in the Malaysian population and subsequently, will reduce the prevalence of NCDs.

## Figures and Tables

**Table 1 foods-11-03791-t001:** The total number of street foods surveyed in this study by food category, districts, and locations of every state.

Regions	Coasts	States	Number of Districts Surveyed	Number of Locations Surveyed	Street Foods Category			Total Street Food by States
					Main Meals	Snacks	Desserts	
West Malaysia	West Coast	Selangor	7	17	792	1000	553	2345
		Federal Territory of Kuala Lumpur	7	14	883	935	328	2146
		Negeri Sembilan	1	6	333	305	164	802
		Melaka	1	5	203	480	156	839
		Johor	8	194	207	220	143	570
		Kedah	3	11	118	64	35	217
		Perlis	2	8	42	47	20	109
		Perak	11	40	90	65	37	192
		Penang	4	34	113	44	43	200
	East Coast	Terengganu	7	10	495	504	396	1395
		Pahang	8	14	362	356	241	959
		Kelantan	2	4	92	102	172	366
East Malaysia	Not applicable	Sabah	4	8	54	55	28	137
		Sarawak	3	15	103	57	83	243
		Total	68	380	3887	4234	2399	10520

**Table 2 foods-11-03791-t002:** The average sodium content in different street food categories and groups.

Food Category	Percentage (%)	Average Sodium Content (mg/100 g)
Snacks	40	433.0 ± 198.1 ^a^
Main meals	37	336.5 ± 148.2 ^a^
Desserts	23	168.0 ± 134.7 ^b^
**Food Group**	**Percentage (%)**	**Average sodium content (mg/100 g)**
Processed foods	8	509.1 ± 144.0 ^a^
Cooked dishes	63	421.8 ± 207.5 ^a^
Local cakes	29	214.0 ± 138.7 ^b^

^a,b^ Different letters indicate significant differences across the column (*p* < 0.05), based on the Games–Howell post hoc test.

**Table 3 foods-11-03791-t003:** Sodium content in processed food prepared as street food.

No	Name of Street Food (*n* = Number of States in Which the Street Food Was Sampled)	Street Food Category	mg Sodium/100 g (Mean ± std dev)	mg Sodium/ Serving (Household Measurement)
1	Fried fish ball (*n* = 3)	Snack	815.0 ± 47.7	1530.9 (6 pieces)
2	Fried chicken with cheese (*n* = 1)	Snack	706.0 ± 24.0	1087.2 (1 piece)
3	Fried crab meatball (*n* = 2)	Snack	690.5 ± 18.4	553.6 (6 pieces)
4	Fried sausage (*n* = 3)	Snack	690.3 ± 74.4	683.4 (3 pieces)
5	Fried chicken ball (*n* = 1)	Snack	605.5 ± 12.0	601.3 (6 pieces)
6	Beef burger (*n* = 1)	Main meal	584.0 ± 2.8	584.0 (1 piece)
7	Fried sausage with cheese (*n* = 1)	Snack	563.0 ± 9.9	802.3 (3 pieces)
8	Chicken nuggets (*n* = 5)	Snack	519.1 ± 73.8	327.0 (3 pieces)
9	Pizza (*n* = 1)	Snack	485.0 ± 12.7	5432.0 (1 whole regular)
10	Chicken burger (*n* = 3)	Main meal	451.8 ± 102.0	682.3 (1 piece)

**Table 4 foods-11-03791-t004:** Sodium content in noodle and rice-based street food with high (>600 mg/ 100 g sample) and medium (120–599 mg/100 g sample) sodium contents.

	High Sodium Content (>600 mg/100 g)			
**No.**	**Name of Street Food (*n* = Number of States in Which the Street Food Was Sampled)**	**Street Food Category**	**mg Sodium/100 g** **(Mean ± Std Dev)**	**mg** **Sodium/** **Serving (Household Measurement)**
1	Fried noodles (*n* = 10)	Main meal	704.2 ± 225.4	2185.5 (1 plate)
2	*Kolo mee* (*n* = 1)	Main meal	625.0 ± 1.4	1152.5 (1 bowl)
	**Medium Sodium Content (120–599 mg/100 g)**			
**No.**	**Name of Street Food (*n* = Number of States in Which the Street Food Was Sampled)**	**Street Food Category**	**mg Sodium/100 g** **(Mean ± Std Dev)**	**mg** **Sodium/** **Serving (Household Measurement)**
1	*Bakso* (*n* = 1)	Main meal	468.5 ± 3.5	2576.8 (1 bowl)
2	Fried *kuey teow* (*n* = 7)	Main meal	437.1 ± 78.6	1477.8 (1 plate)
3	Noodle soup (*n* = 1)	Main meal	431.0 ± 29.7	2249.8 (1 bowl)
4	*Char kuey teow* (*n* = 2)	Main meal	418.3 ± 127.6	1565.5 (1 plate)
5	Fried vermicelli @ fried *mihun* (*n* = 11)	Main meal	398.6 ± 133.6	797.2 (1 plate)
6	*Soto* (*n* = 1)	Main meal	377.0 ± 21.2	3168.7 (1 bowl)
7	Fried rice (*n* = 2)	Main meal	374.8 ± 38.5	749.5 (1 plate)
8	Vermicelli soup @ *mihun sup* (*n* = 3)	Main meal	360.0 ± 78.8	2162.6 (1 bowl)
9	*Nasi lemak* with fried chicken (*n* = 4)	Main meal	349.1 ± 76.2	1431.5 (1 plate)
10	*Nasi tomato* (*n* = 1)	Main meal	345.5 ± 2.1	1935.7 (1 plate)
11	Noodles with gravy (curry/soy sauce) (*n* = 4)	Main meal	336.6 ± 248.4	1830.8 (1 bowl)
12	*Nasi lemak* (*n* = 12)	Main meal	321.7 ± 104.7	643.4 (1 plate)
13	*Kuey teow* soup (*n* = 1)	Main meal	307.0 ± 2.8	2076.4 (1 bowl)
14	*Laksa* (Penang style) (*n* = 5)	Main meal	306.6 ± 93.1	1966.6 (1 bowl)
15	*Nasi minyak* (*n* = 2)	Main meal	304.0 ± 46.0	1128.2 (1 plate)
16	*Nasi kerabu* (*n* = 2)	Main meal	298.5 ± 64.3	916.1 (1 plate)
17	Chicken rice (*n* = 3)	Main meal	298.5 ± 25.1	746.3 (1 plate)
18	*Laksa* (Perak style) (*n* = 1)	Main meal	288.0 ± 0.0	1789.9 (1 bowl)
19	Glutinous rice with *rendang* (meat cooked with spices) (*n* = 1)	Main meal	285.5 ± 17.7	467.7 (1 set)
20	Chicken porridge (*n* = 2)	Main meal	272.5 ± 0.7	452.4 (1 bowl)
21	Rice porridge (*n* = 3)	Main meal	245.5 ± 49.7	407.5 (1 bowl)
22	Glutinous rice with fried fish (*n* = 1)	Main meal	216.0 ± 8.5	335.1 (1 pack)
23	Spaghetti bolognese (*n* = 1)	Main meal	215.5 ± 13.4	1131.4 (1 plate)
24	*Laksam* (*n* = 1)	Main meal	185.5 ± 0.7	519.7 (1 bowl)

**Table 5 foods-11-03791-t005:** Other street foods with medium (120–599 mg/100 g sample) and low (<120 mg/100 g sample) sodium contents.

Street Foods with Medium Sodium Content (120–599 mg/100 g Sample)				
**No.**	**Name of Street Food (*n* = Number of States in Which the Street Food Was Sampled)**	**Street Food Category**	**mg Sodium/100 g** **(Mean ± Std Dev)**	**mg Sodium/Serving (Household** **Measurement)**
1	*Apam balik* with cheese (*n* = 1)	Dessert	586.5 ± 12.0	1343.1 (1 piece)
2	Kebab (*n* = 2)	Main meal	531.5 ± 89.1	616.5 (1 piece)
3	*Kerepek* (*n* = 1)	Snack	512.5 ± 61.5	71.8 (1 piece)
4	Takoyaki (*n* = 4)	Snack	502.4 ± 95.3	926.0 (6 pieces)
5	Fried chicken (*n* = 10)	Snack	464.9 ± 118.3	715.9 (1 piece)
6	Fried chicken (non-meat parts) (*n* = 1)	Snack	461.5 ± 72.8	113.1 (4 small pieces)
7	*Roti john (n = 2)*	Main meal	412.0 ± 59.4	696.3 (3 pieces)
8	Satay (*n* = 2)	Snack	395.3 ± 177.8	296.4 (5 sticks)
9	*Satar* (*n* = 1)	Snack	390.5 ± 0.7	338.3 (3 pieces)
10	*Kuih kacang* (*n* = 1)	Dessert	384.5 ± 16.3	339.9 (3 pieces)
11	*Cakoi* (*n* = 2)	Snack	376.3 ± 223.1	508.4 (3 pieces)
12	Fried *popiah* (*n* = 1)	Snack	350.0 ± 7.1	147.0 (1 piece)
13	*Pulut panggang* (*n* = 1)	Snack	349.5 ± 3.5	174.8 (1 piece)
14	*Putu piring* (*n* = 1)	Dessert	334.5 ± 57.3	184.0 (1 piece)
15	*Kuih bom* (*n* = 1)	Dessert	327.0 ± 4.2	143.9 (1 piece)
16	Curry puff (*n* = 8)	Snack	305.0 ± 87.0	146.5 (1 piece)
16	Banana fritters with cheese (*n* = 1)	Dessert	293.0 ± 4.2	193.4 (3 pieces)
17	*Murtabak* (*n* = 2)	Snack	292.3 ± 188.4	825.6 (1 piece)
18	*Jering rebus* (*n* = 1)	Snack	291.5 ± 0.7	145.8 (4 pieces)
19	*Kuih seri muka* (*n* = 2)	Dessert	271.3 ± 73.2	260.4 (1 piece)
20	French fries with sauce (*n* = 1)	Snack	262.0 ± 7.1	175.5 (1 small serving)
21	*Apam balik telur* (*n* = 1)	Snack	262.0 ± 2.8	600.0 (1 large piece)
22	*Roti canai* (flat bread) (*n* = 2)	Main meal	262.0 ± 154.9	220.1 (1 piece)
23	*Kuih akok* (*n* = 2)	Dessert	259.3 ± 20.2	375.3 (4 pieces)
24	Donut (*n* = 4)	Dessert	254.9 ± 158.1	188.6 (1 piece)
25	*Apam balik* (*n* = 4)	Dessert	253.9 ± 76.1	581.4 (1 large piece)
26	*Kuih cara berlauk ayam* (*n* = 1)	Snack	246.5 ± 3.5	310.6 (4 small pieces)
27	Net crepes *(Roti jala)* (*n* = 2)	Main meal	240.8 ± 1.1	584.9 (1 pack)
28	*Kuih tepung gomak* (*n* = 1)	Snack	235.0 ± 12.7	276.7 (4 pieces)
29	Grilled chicken (small pieces) (*n* = 1)	Snack	229.5 ± 4.9	581.0 (4 small pieces)
30	Corn (savory) (*n* = 2)	Snack	220.9 ± 242.0	1361.3 (1 sheaf)
31	Egg tart (*n* = 1)	Dessert	204.5 ± 9.2	242.9 (1 piece)
32	*Kuih puteri ayu* (*n* = 1)	Dessert	194.0 ± 2.8	276.2 (4 pieces)
33	Banana fritters (*n* = 4)	Dessert	160.3 ± 66.8	105.8 (3 pieces)
34	*Chee cheong fun* (*n* = 1)	Snack	146.0 ± 1.4	826.0 (1 plate)
35	*Beh hua chee* (*n* = 1)	Snack	138.5 ± 12.0	305.4 (3 pieces)
36	*Kuih sagu* (*n* = 1)	Dessert	138.0 ± 0.0	93.8 (1 piece)
37	*Kuih cek mek molek* (*n* = 1)	Dessert	125.5 ± 0.7	128.9 (4 pieces)
38	Popcorn (*n* = 1)	Dessert	121.5 ± 6.4	245.4 (1 container)
**Street Foods with Low Sodium Content (<120 mg/100 g Sample)**				
**No.**	**Name of Street Food (*n* = Number of States in Which the Street Food Was Sampled)**	**Street Food Category**	**mg Sodium/100 g** **(Mean ± Std Dev)**	**mg Sodium/Serving (Household Measurement)**
1	*Kuih keria* (*n* = 1)	Dessert	119.5 ± 2.1	37.0 (1 piece)
2	Cubed rice with peanut gravy *(nasi impit)* (*n* = 1)	Main meal	117.5 ± 0.7	58.8 (8 cubes)
3	*Kuih lepat* (*n* = 2)	Dessert	109.2 ± 44.3	126.6 (1 piece)
4	*Kuih calak kuda* (*n* = 1)	Dessert	104.0 ± 0.0	99.8 (4 pieces)
5	*Kuih buah Melaka* (*n* = 1)	Dessert	89.9 ± 0.6	13.5 (1 piece)
6	*Kuih lapis* (*n* = 2)	Dessert	89.6 ± 93.9	89.6 (1 piece)
7	*Cekodok* (*n* = 1)	Dessert	84.7 ± 33.0	25.4 (3 pieces)
8	Steamed baozi with sweet fillings (*n* = 2)	Dessert	69.1 ± 27.5	34.6 (1 piece)
9	*Peneram* (*n* = 1)	Dessert	59.1 ± 1.6	18.9 (5 small pieces)
10	*Cendol* (*n* = 2)	Dessert	56.9 ± 2.8	431.7 (1 bowl)
11	*Kuih apam* (*n* = 1)	Dessert	56.1 ± 1.3	22.4 (3 pieces)
12	*Kuih jelurut* (*n* = 1)	Dessert	49.8 ± 2.1	101.6 (4 pieces)
13	*Tau fu fa* (*n* = 1)	Dessert	6.4 ± 0.4	23.6 (1 container)
14	*Kuih penjaram* (*n* = 1)	Dessert	5.3 ± 0.1	9.1 (4 pieces)
15	*Kuih angku* (*n* = 1)	Dessert	5.2 ± 0.6	14.1 (4 pieces)
16	*Putu* (*n* = 1)	Main meal	4.1 ± 0.2	7.2 (1 piece)
17	*Sianglag* (*n* = 1)	Main meal	3.9 ± 0.7	4.9 (1 pack)

## Data Availability

Data is contained within the article or Appendix A.

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
