# Peer review of "Street Food in Malaysia: What Are the Sodium Levels?"

_foods, 2022, doi:10.3390/foods11233791_

Round 1

Reviewer 1 Report

The manuscript titled “ Street food in Malaysia: What are the sodium levels?” was reviewed for consideration in foods. The topic is innovative as well as very interesting. However, following there are some comments needs to understand and where necessary should be included in manuscript.

During introduction

It should also be worth mentioning if we suggest using potassium chloride salt in daily life, because it will maintain taste as well as not take part in sodium cycle in body.

Line 79: Could you provide survey form, and include it as supplementary file too, in order to better understand the frequency data???

Line 116-18: What is the classification of sodium in Malaysia?

Line 127-129: The ANOVA was used for significant difference evaluation, however how the difference between different food category was found? LSD or Tukey’s range test? Please mention this specific information.

Methodology

There is not mentioning anything about the condition of ICP instrument, and LOD or LOQ data regarding sodium, standard curve etc., which is the backbone of reliable, accurate and precise results.

Table 1: I think Kuala Lumpur is not state, its included in Selangor state. Please double check.

Conclusion

This part needs to be elaboration, please rewrite it with suggestions and health effects of using high amount of sodium in our daily life.  

Reviewer 2 Report

The MS is interesting but requires some modifications:

1. Abstract should be re-written as per the journal guideline, it should not be segregated into different parts. The aim and novelty should be reflected in the abstract portion.

2. Is there any other studies available on the street foods of other countries? In the introduction, the previous instances where street foods from different countries should be covered

3. In table 3-5 the superscripted letters for the significant difference are missing.

4. Flowchart of the detailed research work needs to be incorporated.

5. Survey questionary needs to be included in the MS.

6. Conclusion part needs to be rewritten.

7. More recent and relevant research should be included in the MS.

Reviewer 3 Report

General comments to the authors

 The topic of this research article is of interest to the readers of the journal and timely relevant, both considering Malaysia’s and the global public health scenarios. The research question is well designed: to identify the most frequently available street foods in Malaysia and determine its sodium content.

The article lacks important descriptions in the methodology and results, to increase transparency and to allow drawing conclusion about the representativeness of the study. Several recommendations for changes in the results, discussion and conclusion sections are additionally suggested. For example, the discussion should consider the comparison of these findings with other findings from the literature, which is not the case. As it is, the discussion is a repetition of the results’ section. In addition, the conclusions should be broader, related to the main relevant findings and provide a summary of the study implications. I suggest major changes to clarify the issues raised (comments below). The article lacks revision by and English native speaker.

Specific comments to the authors

Abstract

Line 20-22: These 2 sentences are not in accordance. Is the level of sodium in street food (SF) samples known or unknown? Please clarify.

The presentation of results should be revised according to the comments in the results’ section.

Introduction

Line 37-38: the reported number of NCD deaths is global? In the region? The reference is from the European Commission.

Line 38: “In Malaysia (…)” – please revise the sentence, as it is not clear as it is.

The first paragraph is confusing. I suggest revision.

Refs. Line 57 – there is a key systematic review on this topic, that provides this information from the global perspective. Doi: 10.1017/S1368980013001158.

The introduction does not clarify why the focus of the current article on street food. A few questions:

- Are there any national studies reporting the relevance of street food for the diet of the population, for example? Why selecting street food outlets and not other types of food establishments such as restaurants?

- Also, did the authors consider street food as only unpackaged foods?

Methods

- Please provide a first paragraph contextualizing the research. Important information to include: how is the country organized/how many states, socioeconomic background, urbanization level, legislation on street food, or sodium content, how is the SF environment organized (e.g. in markets, or randomly throughout the cities/villages ..)

- Please explicit the reason for conducting the study in all states. Were both urban and rural contexts included, or only urban? Also, add the period of study implementation (data + sampling collection).

Lines 68-69 – You should provide the reference for the street food definition used. Was it FAO/WHO?

Lines 72-75: Was this definition based on some recommended criteria? If so, the reference should be provided.

Line 76: why “still” operating? Have these vending sites been identified from any “official” registry? Please clarify.

Line 79: what is a survey form? Were the SF vendors interviewed? Or was the data collected through observation only?

Lines 79-86: What was the reasoning for selecting 4 or 5 SF establishments in each state? Is this representative of the SF offer in that state?

Lines 81-83: remove the sentence. The information is duplicated, and it is more appropriately described in the statistical analysis subsection.

Lines 84-86: Why selecting the top 15 frequently available SF? Why did you consider only these 3 food categories? Did you only identify SF pertaining to these categories? According to the global evidence on SF, additional categories of SF were expected to have been identified.

Line 96: I do not understand the criterion ii). Wasn’t phase II conducted immediately after phase I? Or was there a long-time gap between these 2 phases? Please clarify.

Line 104: why did you weight the purchased samples together with the packaging?

Lines 106-107: you should have explained previously that the food samples were purchased from two different stalls. It is not clear how many samples were purchased, in total, nor to how many different foods do they correspond. Section 2.2.1 should be entirely revised and the sampling process better clarified.

Line 116: a reference is missing.

Section 2.3: please specify the statistical tests used. The description is not clear.

Lines 124-125: “methods of street food preparation were also determined”. Was this a statistical procedure? I believe this information should figure in subsection 2.1.

Was this study approved by an Ethics Committee? This information should be provided in this section.

Results

Please remove the first sentence.

Context for the second sentence should be provided in the methods, for better interpretation of this finding.

Line 134: Is Johor a district, state…?

Line 136: n=10520 types of street food. Are these all different foods? Or some of these foods are the same? Did you select 15 SF to be collected/analysed, out of a pool of 10520? This raises concerns regarding the representativeness of the sample, regarding the global SF offer in the country

Lines 138-140: The sum of the referred cooking methods is 72.3%. What were the remaining cooking methods, not described?

Line 145: n=210, or 420? 2 samples of each food were described previously. Please clarify in section 2.2.1

Line 147: processed food was not mentioned earlier in the manuscript, which should be clarified. Is this a food category? Or a different classification?

Lines 149-150: isn’t the number of SF collected in each state dependent on the size of the state (e.g. higher number of districts)? The sampling procedure should clarify this. And, if so, wouldn’t this result be expected? Is it that relevant for the aim of the study?

Line 160: “local cakes” – did you distinguish local and “western” recipes? If so, you should also clarify in the methodology.

Tables 3-5: please add a column with the number of samples of each street food used to determine the mean +- std dev

Section 3.3. Was the frequency of each food category similar across coasts/regions?

Paragraphs 1 and 2 seem to present contradictory findings. Please revise.

Discussion

Please revise the entire section. The first paragraph should present a summary of the main finings and the remaining should address comparisons with previous literature on the topic and implications of these findings (e.g., for current policies, etc.).

Conclusion

Line 305-306: Isn’t this finding obvious? You should state the important and relevant findings here, and a summary of the study implications.

Did the findings of the study answer the research question?

Supplementary material: please provide photographs or descriptions of the SF purchased/analysed, which will allow a better interpretation of the findings for all potential readers.

Round 2

Reviewer 1 Report

The revised version has been updated according to comments raised.

Reviewer 2 Report

Quality of the MS has been improved